# Clinical Characteristics, Exercise Capacity and Pulmonary Function in Post-COVID-19 Competitive Athletes

**DOI:** 10.3390/jcm10143053

**Published:** 2021-07-09

**Authors:** Klara Komici, Antonio Bianco, Fabio Perrotta, Antonio Dello Iacono, Leonardo Bencivenga, Vito D’Agnano, Aldo Rocca, Andrea Bianco, Giuseppe Rengo, Germano Guerra

**Affiliations:** 1Department of Medicine and Health Sciences, University of Molise, 86100 Campobasso, Italy; antoniodottorbianco@gmail.com (A.B.); vito.dagnano94@gmail.com (V.D.); aldo.rocca@unimol.it (A.R.); germano.guerra@unimol.it (G.G.); 2Exercise and Sports Medicine Unit, “Antonio Cardarelli Hospital”, 86100 Campobasso, Italy; 3UOC Pneumologia AORN Sant’Anna e San Sebastiano, 81100 Caserta, Italy; Dottfabioperrotta@gmail.com; 4School of Health and Life Sciences, University of the West of Scotland, Hamilton G72 0LH, UK; Antonio.DelloIacono@uws.ac.uk; 5Department of Translational Medical Sciences, University of Naples “Federico II”, 80131 Naples, Italy; leonardobencivenga@gmail.com (L.B.); giuseppe.rengo@unina.it (G.R.); 6Department of Advanced Biomedical Sciences, University of Naples “Federico II”, 80131 Naples, Italy; 7Department of Translational Medical Sciences, University of Campania “L. Vanvitelli”, 80138 Naples, Italy; andrea.bianco@unicampania.it; 8Istituti Clinici Scientifici Maugeri SpA Società Benefit (ICS Maugeri SpA SB), 82037 Telese Terme, Italy

**Keywords:** COVID-19, SARS-CoV-19, physical exercise, CPET

## Abstract

Background: Limited evidence exists regarding adverse modifications affecting cardiovascular and pulmonary function in physical active adults affected by COVID-19, especially in athletic populations. We aimed to describe the clinical presentation of COVID-19 in a cohort of competitive athletes, as well as spirometry and echocardiography findings and cardio-respiratory performance during exercise. Methods: Twenty-four competitive athletes with COVID-19 were recruited for this study after ending self-isolation and confirmation of negative laboratory results. All athletes underwent clinical evaluation, spirometry, echocardiography and cardiopulmonary exercise testing (CPET). These data were compared to a group of healthy control athletes. Results: Anosmia was the most frequent symptom present in 70.83% patients, followed by myalgia, fatigue and ageusia. The most frequent persisting symptoms were anosmia 11 (45.83%) and ageusia 8 (33.33%). Compared to controls, COVID-19 patients presented lower FEV1%: 97.5 (91.5–108) vs. 109 (106–116) *p* = 0.007. Peak Oxygen Uptake (VO_2_) in COVID-19 patients was 50.1 (47.7–51.65) vs. 49 (44.2–52.6) in controls (*p* = 0.618). Conclusions: Reduced exercise capacity was not identified and pulmonary and cardiovascular function are not impaired during early recovery phase in a population of physical active adults except FEV1 reduction.

## 1. Introduction

The Coronavirus disease 2019 (COVID-19), caused by the novel severe acute respiratory syndrome coronavirus 2 (SARS-CoV-2), has affected millions of people globally [1]. The most commonly described symptoms are fever, cough, shortness of breath, anosmia, ageusia, fatigue, myalgia, and nausea/vomiting or diarrhea [2]. Complications of COVID-19 in hospitalized patients include respiratory failure, thromboembolic events, myocarditis, arrhythmias and hemodynamic instability. Young and fit individuals with COVID-19 seem to generally suffer from mild to moderate symptoms [3], which are limited to the upper respiratory tract, without requiring hospitalization or additional laboratory, chest X-ray or lung computer tomography (CT) examination. Importantly, a recent study performed on athletes with previous asymptomatic to mild COVID-19 did not reveal significant anomalies in laboratory tests, echocardiography and exercise test monitoring [4]. However, several reports have described prolonged impairment in lung function in post-COVID-19 patients [5,6], with a significant percentage of patients after rehabilitation programs, which still present exertional dyspnea. The altered lung function, observed in some post-COVID-19 patients [6] and persistent ground glass opacity together with a lack of improvement in respiratory function for up to 12 months [7], raise concerns regarding long-term impairment of respiratory system and physical performance. At present, several information regarding myocarditis in young athletes after COVID-19 disease has been described with heterogeneous prevalence, ranging from 0–15% [8], indicating the necessity of further data and standardized screening protocols before a safe return to play. In addition, post-viral myocarditis has been documented as a cause of sudden cardiac death in young competitive athletes [9], and experimental animal models of myocarditis have shown that physical exertion may have fatal consequences [10]. However, the evidence on the impact of COVID-19 in athletes remain poor, and actually no accurate definition is available regarding clinical presentation and impact of the disease on physical performance. Therefore, in the present study we aimed to: (a) Describe COVID-19 clinical presentation in a cohort of competitive athletes; (b) evaluate in competitive athletes the impact of SARS-CoV-2 infection on cardio-respiratory performance at rest and during exercise. 

## 2. Materials and Methods

### 2.1. Study Population

Competitive athletes affected by COVID-19 were considered for enrollment in this study. Inclusion criteria were: (a) Age ≥ 18 years and ≤35; (b) positive testing to SARS-CoV-2 by reverse transcription polymerase chain reaction–based (rt-PCR) SARS-CoV-2 RNA from nasopharynx swab; (c) negative rt-PCR SARS-CoV-2 RNA performed 10 days after the first positive nasopharynx, and end of self-isolation period, as indicated by National Government Recommendations (Italian Ministry of Health 12 October 2020 Normative); (d) willingness to participate in this study. Physical examination, weight, height, Body Mass Index (BMI), resting electrocardiogram (ECG), resting systolic and diastolic blood pressure (SBP, DBP) were registered for all patients. Individual records regarding symptoms presentation, their duration in days and persistence if at the time of the study commencement any were also collected. In addition, previous medical conditions, such as asthma, allergy, hypertension, diabetes, dyslipidemia, arrhythmia, heart valve diseases, familiarity for sudden cardiac death and cardiovascular risk factors were collected. A group of competitive athletes following previous physical capacity evaluation after summer holidays and before starting training program tested negative for COVID-19 was recruited as a matched control group. Post-Covid-19 athletes and controls were soccer players. Patients with positive nasopharynx swabs, non-competitive athletes and recovered patients from more than 30 days were excluded from the study. 

This study was approved by the Institutional Review Board of Department of Medicine and Health Sciences, University of Molise protocol number 07/2021, and conducted in compliance with the Declaration of Helsinki. All participants gave written informed consent. 

### 2.2. Spirometry 

Spirometry was performed in accordance to recommended standards [11] with the subjects seated, on the same day immediately before the exercise test. Forced Vital Capacity (FVC), Forced Expiratory Volume in one second (FEV1), FEV1/FVC ratio and Forced Expiratory Flow at rates 25–75% were measured using a clinical spirometer (Sensormedics Viasys Carefusion Vmax Encore 22). The spirometry measurements were also expressed as percentages of theoretical values using reference values normative scores determined through prediction equations, as described elsewhere [12,13].

### 2.3. Echocardiography

A standard trans-thoracic echocardiography was performed using a Philips iU22 ultrasound system with cardiac sector transducer sampling at 1–5 MHz. All measurements were performed according to the European Society of Cardiology Recommendations for Chamber Quantification [14]. Left Ventricular (LV) size end-diastolic and end-systolic diameter, volume and wall thickness were measured. Global and regional LV functions were evaluated and LV Ejection Fraction (LVEF) was calculated from apical four- and two-chamber views using Simpson’s biplane method. Right ventricle size kinetics and function, such as the tricuspid annular plane systolic excursion (TAPSE); fractional area of contraction (FAC), and; presence of pericardial effusion were evaluated. Peak flow velocity during the early diastolic filling phase (E) and during the atrial contraction (A), as well as the deceleration time (DT) were measured. For each Doppler-based measurement, estimates were obtained from 3 cardiac cycles and averaged.

### 2.4. CPET 

Ventilation and gas exchange variables were measured continuously using a breath-by-breath gas analysis system Omnia Quark CPET Cosmed 2019, which was calibrated according to the instructions of the manufacturer. Participants were fitted and familiarized to a two-way breathing Hans Rudolph 7450 series V2mask and headgear before stepping on a motorized treadmill Cosmed T150med and starting the test. A 12 lead ECG recording system Quark T12X wireless12-lead ECG was used continually. Arterial blood pressure was measured at the end of each stage of the test. Non-invasive saturation of peripheral oxygen at rest (SpO_2_%), peak effort and at the end of recovery were assessed. To evaluate significant desaturation during exercise test the difference between rest and peak effort oxygen saturation was calculated. All patients performed an incremental exercise test beginning with 8 km/h speed with stepwise increases of 1 km/h every minute. After reaching a speed of 14 km/h, the inclination was increased by 1% every minute. Patients were encouraged to continue the exercise test until a maximal effort attainment indicated by: (a) Failure of oxygen uptake or heart rate (HR) increase with further increase in work rate; (b) peak respiratory exchange ratio (RER) ≥ 1.10; (c)) rating of perceived exertion ≥ 8 (on the 10-point Borg scale) [15,16]. Peak VO_2_ was recorded as the highest averaged value across 10–30 s during exercise phase. The first ventilatory threshold (1st VT) estimated by the V-slope or respiratory equivalents methods, Ventilation (VE)/Volume of exhaled carbon dioxide (VCO_2_) slope were evaluated as determined by Wasserman et al. [17] and Clinical Recommendations for Cardiopulmonary Exercise Testing Data Assessment in Specific Patient Populations [18].

### 2.5. Statistical Analysis 

Data are expressed as median and interquartile range (IQR) for continue variables and as absolute frequencies and percentage for categorical variables. Since the normality distribution of different variables could not be assumed as evaluated by Shapiro-Wilk test, univariate analysis Wilcoxon rank-sum (Mann-Whitney) test was performed for continuous variables. The association between categorical variables was assessed by Chi square test. Statistical significance was at *p* ≤ 0.05. Statistical analyses were performed with STATA SE 16.1 software.

## 3. Results

### 3.1. Clinical and Instrumental Findings in COVID-19 Athletes 

Twenty-four post-COVID-19 athletes represented the final study population. Two patients with negative nasopharynx swabs, performed one month previously, were excluded. Detailed descriptions of COVID-19 clinical presentation, medical history, duration of main symptoms, persistence of symptoms, clinical and instrumental evaluations are shown in Table 1. The most frequent symptoms reported were anosmia presented by 17 (70.83%) subjects followed by myalgia, fatigue and ageusia, observed in 16 (66.67%), 15 (62.5%) and 15 (62.5%) patients, respectively. Less common symptoms were: fever and rhinitis (*n* = 12; 50%), cough (*n* = 11; 45.83%), pharyngodynia (*n* = 9; 37.5%) shortness of breath (*n* = 5; 20.83%) and headache (*n* = 3; 12.5%). Diarrhea was present in 4 patients (16.67%), while only 1 patient (4.17%) suffered from nausea. The mean symptom duration ranged from 2 to 5 days, while only two patients were completely asymptomatic. Figure 1 depicts frequency distribution of symptoms and their persistence at the time of the study enrollment. Shortness of breath, fever, headache or gastro-intestinal disorders were not persisting symptoms in the study population. The most frequent persisting symptoms were anosmia (*n* = 11; 45.83%) and ageusia (*n* = 8; 33.33%). Sporadic cough was present in 4 patients (16.67%), rhinitis in 5 patients (20.83%) and myalgia in 2 patients (8.33%). Five patients (20.83%) did not present symptoms’ persistence. Spirometry parameters revealed a reduction in FEF 25–75% in 6 (25%) patients without significant alternation in other parameters. Clinical examination, EKG and echocardiography did not show any significant abnormality, with the exception of physical activity-induced modification in cardiac function and structure. EKG monitoring, during CPET, reported isolated Premature Ventricular Beats (PVB) and Premature Supraventricular Beats (PSVB) in two patients and T wave inversion without chest pain or other symptoms in one patient. These alterations were already described in previous tests, and were performed before the COVID-19 pandemic. Cardiac magnetic resonance imaging (CMR) together with CT coronary-angiography did not report any pathological finding. Predicted Peak VO_2_% was more than 100% in all patients except for one patient. However, none of these patients showed ventilatory inefficiency or significant desaturation, but only muscles discomfort. In all patients, the VE/VCO_2_ slope was less than 30 and no significant reduction of SpO_2_ during exercise was registered. None of the patient presented Ventilatory Oscillations (VO) during exercise. Table 2 reports the main CPET data in COVID-19 patients.

### 3.2. Comparison between COVID-19 and Healthy Controls (HC)

No statistically significant differences between post-COVID-19 athletes and SARS- CoV-2-negative players were found for age, BMI, previous medical conditions and ECG. Echocardiography data did not show differences in LV systolic or diastolic function and structure. Pulmonary artery pressure (PAP) was higher in post-COVID-19 patients compared to controls (24 {21–25.5} vs. 18 {17–22}; *p* = 0.017). Other data as tricuspid annular plane systolic excursion (TAPSE), TAPSE/PAP ratio and Fractional Area Change (FAC) did not differ significantly between the two groups (23 {20.8–25} vs. 21.4 {21–24}; *p* = 0.519; 0.96 {0.84–1.14} vs. 1.15 {0.95–1.33}; *p* = 0.179; 50.75 {45.55–55.65} vs. 52 {48.6–54.3}; *p* = 0.582, respectively). Spirometry data revealed no differences regarding almost all parameters, with the exception of FEV1% of theoretical value, with lower values in post-COVID-19 athletes (97.5 {91.5–108} vs. 109 {106–116}; *p* = 0.007). Peak VO_2_ in post-COVID-19 subjects was 50.1 (47.7–51.65) vs. 49 (44.2–52.6) in controls (*p* = 0.618). No statistical difference was found in resting and peak exercise HR, BP and VE. Furthermore, peak VCO_2_, peak Oxygen pulse, peak RER, VE/VCO2 slope and lowest VE/VCO2 were not different between post-COVID-19 athletes and controls. A trend to reduction in VO_2_ at 1st VT expressed as % of peak VO_2_ was observed in the post-COVID-19 group compared to controls (74.07% {71.6–78.6}vs. 79.04% {76.04–80.01}; *p* = 0.082. Echocardiographic, spirometry and CPET data are reported in Table 3.

## 4. Discussion

The main findings of the present study, included; (a) the clinical presentation of SARS-CoV-2 infection in competitive athletes is mainly characterized by anosmia, ageusia, myalgia and fatigue with a frequency around 60–70%. Fever and upper respiratory tract symptoms were present in 40–50% of the subjects. Other symptoms as shortness of breath or gastro-intestinal symptoms were less common; (b) cardio-respiratory function was not significantly altered in post-COVID-19 athletes both, at rest and during exercise; (c) post-COVID-19 athletes showed a reduction in FEV1% of theoretical value compared to SARS- CoV-2-negative competitive players, despite that ventilatory efficiency and overall performance were not impaired.

The most frequent COVID-19 symptoms, described in non-hospitalized adult patients, include fever, fatigue and cough [19]. Anosmia and ageusia are reported more often than in other viral infections [20]. However, in adult competitive athletes, we found that anosmia and myalgia are most frequently reported. Fever and dry cough are present in up to 90% of hospitalized patients [2]. Whereas, anosmia, fever and myalgia were the only symptoms associated with SARS-CoV-2 positivity among healthcare workers [21].

Physical fitness and frequent aerobic training have been demonstrated to reduce the frequency, severity and symptomatology of upper respiratory tract infections [22]. Although the role of exercise in specific immune mechanism is not well-established, it has been reported that exercise modifies the number of circulating lymphocytes and the release of stress hormones, and in particular, cortisol, which may influence the activity of anti-inflammatory cytokines [23]. In addition, experimental models report that lung macrophages mediates the beneficial effects of exercise regarding response to infections [24]. From the other side, SARS-CoV-2 infection impairs lymphopoiesis, and the thickening of alveolar wall with macrophages and mononuclear cells have been found in lungs of COVID-19 patients [25]. Furthermore, obesity has emerged as a risk factor for COVID-19 severity and negative outcome [26], and the protective role of exercise in weight control in COVID-19 presentation and consequences should be further investigated.

In our cohort of competitive athletes’ spirometry, the findings did not reveal modifications related to SARS-CoV-2 infection with the only exception of reduced FEV1%. It has been reported that 11% of patients with severe COVID-19, had reduced FVC [7], while FEV1% was significantly lower in patients after severe or critical COVID-19 compared to patients presenting mild to moderate disease [27]. In a recent study, Gervasi and colleagues [4] found a reduction of PEF in professional athletes recovering from previous mild-to-moderate symptomatic SARS-CoV-19 infection. However, the authors could not clarify whether the observed clinical outcomes resulted from the disease condition or as a consequence of the forced detraining period. Conversely, we did not found any difference in FEF25–75%. Although FEF25–75% measurements could theoretically have higher sensitivity in detecting small airway disease. To date, there has been no reported evidence of changes in subjects suffering from SARS-CoV-2 infection [28]. A study performed on patients with suspected or known lung disease concluded that the contributive role of FEF25–75% measurements, in addition to FEV1, FVC and FEV1/FVC ratio for effective clinical decision making is trivial. [29]. However, it has been reported that in the asthmatic population, changes in FEV1 of more than 20% in short-term trials are confident to a clinically significant change [30]. In our study, we did not observe any difference in FEF among post-COVID-19 and SARS-CoV-2 negative athletes. Whereas, COVID-19 was associated with a reduction in FEV1%. Data regarding minimal clinically important difference regarding pulmonary parameters in post-Covid-19 are lacking and future studies are needed to explore whether these variations have a clinical significance.

We underline that none of the patients presented sinus tachycardia however an implication of autonomic nervous system imbalance has been suggested in “long COVID” [31]. The effects of SARS-CoV-2 infection on cardiac function in competitive athletes are not fully clarified. A previous report investigating cardiac function through cardiac magnetic resonance (CMR) has reported results that are suggestive of myocarditis [32]. However this finding remains debated. In a cohort of 145 subjects, Starekova and colleagues [3] observed that only 2 athletes (1.4%) showed CMR evidence of acute inflammation. Furthermore, in a cohort performed on 1597 after COVID-19 infection, CMR screening detected 37 subjects (2.3%) with clinical and subclinical myocarditis [33], and recent data from a multi-centered prospective study reported that the prevalence of myocarditis following SARS-CoV-2 infection is about 0–5–3% and no adverse cardiac events were registered during a median follow-up of 113 days in young competitive athletes with RMN signs of myocarditis [34]. Other studies evaluating symptom persistence in adults with mild COVID-19 disease reported that, during three-to-six month follow-up palpitations, chest pain and dyspnea were present, respectively in 9, 5 and 30% of adults [35]. Another study with 1946 patients who suffered out-of-hospital cardiac arrest and 1080 in hospital cardiac arrest had a 2.3 to 3.4-fold risk of dying within one month, and COVID-19 was involved in at least 10% of out of hospital cardiac arrest [36]. Post-mortem studies of athletes who died from sudden cardiac death myocarditis was diagnosed in up to 8% [37]. Moreover, only in 21.6% of sudden cardiac deaths related to myocarditis among young athletes was characterized by typical symptoms as chest pain, dyspnea and palpitations [38]. Therefore, short- and long-term outcome data are strongly required to clarify the implications of COVID-19 disease on cardiac events. Compared to HC, a higher PAP level in COVID-19 patients was found. It should be mentioned that none of the patients presented pathologically elevated PAP and other data regarding RV function as TAPSE, TAPSE/PAP ratio, F.A.C or even pulmonary acceleration time were not different. However, in hospitalized non-intensive care unit patients the prevalence of pulmonary hypertension was 12%, a finding that warrants further investigations to establish its causal-relationships with the COVID-19 disease [39]. CPET examination did not reveal significant modifications regarding exercise limitation, ventilatory efficiency, respiratory reserve and oxygen pulse, indicating no impairments were present in pulmonary or cardiac function, at least in adult competitive athletes at early post COVID-19 recovery. This is not surprising, given that exercise-induced ventilatory limitation was not detected, even in post-SARS patients requiring hospitalization [40]. Despite limited evidence regarding COVID-19 sequelae on respiratory and physical function, few reports suggest impairment during short-term follow-up. Moreover, a recent study described that patients who experienced severe-to-critical COVID-19 disease are characterized by a reduction in pulmonary function as measured by diffusing capacity of the lung for carbon monoxide (DLCO), radiological findings of ground glass opacities with mosaic attenuation, seen up-to-four months during follow-up, which can lead to ventilation/perfusion mismatching and contribute to physical impairment and hypoxemia during exercise [27]. Of interest, we noticed that the first VT expressed as percentage of peakVO_2_ was lower than in HC. Modification of first VT has been described in detrained subjects [17]. Although, the underpinning molecular mechanisms triggered by SARS-CoV-19 infection and the consequences on modifying the ventilatory drive response to excessive lactate accumulation is unknown yet.

This study presents a few limitation worthy of consideration. First, it was performed in a single center on a relatively small number of patients, which limits the possibility of generalizing the observed findings to other populations. Second, while the main symptoms were self-reported during interviews performed briefly after patients’ recovery, no indication of their severity was obtained. Finally, no instrumental examinations were administered for the recruited patients before they contracted COVID-19, which precludes any causal relationships assumption about the effects SARS-CoV-19 infection on the cardiorespiratory and functional outcomes. However, the main anthropometric, clinical history and athletic characteristics (e.g., age, BMI, comorbidities, training background) were homogenous between the COVID-19 group and the healthy control group. Moreover, the control group was tested immediately prior to the start of the competitive season, whereby it can be assumed that this cohort is representative of competitive athletes and used for comparative analysis. Finally, in this study, we did not provide follow-up data regarding cardio-respiratory consequences of SARS-CoV-19 infection in competitive athletes, and long-term implications could not be excluded. However, baseline examinations and CPET monitoring excluded the presence of complex arrythmias or other conditions of clinical significance.

## 5. Conclusions

The COVID-19 disease seems to affect competitive athletes mainly with clinical symptoms, such as anosmia, ageusia myalgia and fatigue. Persisting symptoms, such as anosmia, ageusia, sporadic cough and myalgia may also present in these athletes. Reduced exercise capacity was not identified, and pulmonary and cardiovascular function were not found to be impaired during the early recovery phase in a population of physically active adults, except FEV1 reduction.

## Figures and Tables

**Figure 1 jcm-10-03053-f001:**
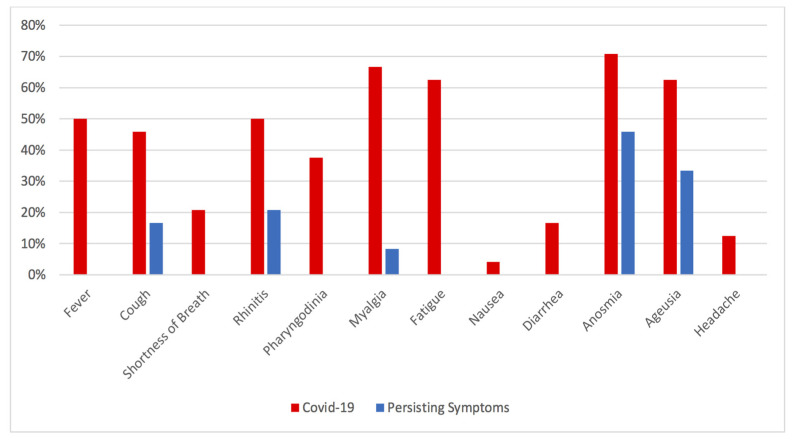
Clinical Presentation of COVID-19 in athletes and Persisting Symptoms.

**Table 1 jcm-10-03053-t001:** Description of clinical and instrumental findings in COVID-19 athletes.

ID	COVID-19 Related Symptoms	Main Symptoms Duration	Persisting Symptoms	Medical History	General Clinical Examination	Resting ECG	Spirometry	Echocardiography
**1**	Fatigue, Myalgia, Rhinitis	4 days	Mild Myalgia	Isolated PVB	Normal	Normal	FEF 25–75% Reduction	Mild Mitral Insufficiency
**2**	Fever, Cough, Anosmia, Ageusia	3 days	Anosmia, Ageusia	Asthma	Normal	Sinus Bradycardia, Incomplete RBB	Normal	Mild Tricuspid Insufficiency
**3**	Fever, Anosmia, Myalgia	5 days	Anosmia	None	Normal	Incomplete RBB	FEF 25–75% Reduction	Normal
**4**	Fever, Cough, Myalgia, Fatigue, Anosmia, Ageusia	3 days	Sporadic Cough	T inversion during exercise test	Normal	Normal	Normal	Mild Tricuspid Insufficiency
**5**	Myalgia, Fatigue	3 days	None	Pollen Allergy	Normal	Normal	FEF 25–75% Reduction	Normal
**6**	Rhinitis, Pharyngodynia	3 days	Rhinitis	None	Normal	Normal	Normal	Normal
**7**	None	N/A	N/A	None	Normal	Sinus Bradycardia grade AV Block, Incomplete RBB	Normal	Normal
**8**	Pharyngodynia, Anosmia, Ageusia	2 days	Anosmia	Hypothyroidism	Normal	Sinus Bradycardia	Normal	Normal
**9**	Fever, Cough, Rhintis, Myalgia, Fatigue, Anosmia, Ageusia	4 days	Rhinitis, Anosmia, Ageusia	None	Normal	incomplete RBB	Normal	Normal
**10**	Disosmia, Ageusia	3 days	None	None	Normal	Normal	FEF 25–75% Reduction	Normal
**11**	Fever, Dispnea, Rhinitis, Pharyngodynia, Anosmia, Ageusia, Myalgia, Fatigue	5 days	Rhinitis, Anosmia, Ageusia	Hypothyroidism, Smoking	Normal	Normal	Normal	Mild Mitral Insufficiency
**12**	Fever, Rhinitis, Myalgia, Fatigue	3 days	Rhinitis	None	Normal	Incomplete RBBB	Normal	Normal
**13**	Myalgia, Fatigue	3 days	None	PFO	2/6 sistolic murmur	Incomplete RBBB	Normal	PFO, Mild Mitral Insufficiency
**14**	Fever, Cough, Pharyngodynia, Myalgia, Fatigue, Anosmia, Ageusia	4 days	Anosmia, Ageusia	none	normal	normal	Normal	Normal
**15**	Cough, Rhintis, Shortness of Breath, Myalgia, Fatigue, Anosmia	3 days	Anosmia	Isolated PVB, Asthma, Pollen Allergia	Normal	Normal	Normal	Normal
**16**	None	N/A	N/A	None	Normal	Sinus Bradycardia	Normal	Normal
**17**	Fever, Cough, Shortness of Breath, Pharyngodynia, Cefalea, Myalgia, Fatigue, Anosmia, Ageusia	5 days	Mild Myalgia	Pollen Allergia	Normal	Normal	Normal	Normal
**18**	Fever, Cough, Rhintis, Dispnea, Faringodinia, Headache, Fatigue, Anosmia, Ageusia, Nause, Diarrhea	6 days	Anosmia, Ageusia, Sporadic Cough	None	Normal	Sinus Bradycardia, Incomplete RBB	Normal	Normal
**19**	Cough, Rhintis, Pharyngodynia, Anosmia, Ageusia	5 days	Anosmia, Ageusia	None	Normal	Sinus Bradycardia	Normal	Mild Mitral Insufficiency, Mild Tricuspid Insufficiency
**20**	Cough, Anosmia, Ageusia	4 days	Sporadic Cough	Asthma, Pollen Allergia	Normal	Incomplete RBB	FEF 25–75% Reduction	Mild Mitral Insufficiency
**21**	Cough, Rhintis, Myalgia, Fatigue, Anosmia, Ageusia	4 days	Rhinitis	none	Normal	Normal	Normal	Normal
**22**	Fever, Dispnea, Pharyngodynia, Rhitntis, Myalgia, Fatigue, Anosmia, Ageusia, Diarrhea	4 days	Anosmia, Ageusia	Pollen Allergia	Normal	Sinus Bradycardia	Normal	Normal
**23**	Fever, Rhinitis, Pharyngodynia, Headache, Myalgia, Fatigue, Diarrhea, Anosmia, Ageusia	5 days	Anosmia, Ageusia	Aortic Insuficiency, Smoking	2/6 diastolic murmur	Normal	Normal	Mild Aortic Insufficiency
**24**	Fever, Cough, Rhinitis, Myalgia, Fatigue, Anosmia, Ageusia, Diarrhea	4 days	Sporadic Cough	Smoking	Normal	Normal	FEF 25–75% Reduction	Normal

PVB: Premature Ventricular Beats; PSVB: Premature Supraventricular Beats; RBBB: Right Bundle Branch Block; FEF 25–75%: mean of Flow Expiratory Flow at 25–75%.

**Table 2 jcm-10-03053-t002:** Description of CPET in COVID-19 athletes.

ID	Peak VO_2_% Predicted	Peak RER	1st VT %VO_2_peak	VE/VCO_2_ Slope	Rest SpO_2_	Peak SpO_2_	Peak VE/MVV	Arrythmias	ST-T Anomalies	Systolic and Diastolic Blood Pressure Profile
**1**	>100	0.93	73.7	<30	97	95	<0.8	Isolated PVB and PSVB.	None	Normal
**2**	>100	1.05	82.1	<30	98	96	<0.8	None	None	Normal
**3**	>100	1.14	70.5	<30	98	96	<0.8	None	None	Normal
**4**	>100	1.06	82.8	<30	98	97	<0.8	Isolated PVB	T inversion in V4-V6.	Normal
**5**	<100 (87)	1.08	82.5	<30	98	97	<0.8	None	None	Normal
**6**	>100	1.07	72.6	<30	98	95	<0.8	Isolated PSVB.	None	Normal
**7**	>100	1.05	80.3	<30	98	96	<0.8	None	None	Normal
**8**	>100	1.03	70.9	<30	98	95	<0.8	None	None	Normal
**9**	>100	1.11	64.5	<30	98	95	<0.8	None	None	Normal
**10**	>100	1.17	74.5	<30	98	96	<0.8	None	None	Normal
**11**	>100	1.14	79	<30	97	95	<0.8	None	None	Normal
**12**	>100	1.05	74.2	<30	99	96	<0.8	None	None	Normal
**13**	>100	1.11	73.9	<30	97	95	<0.8	None	None	Normal
**14**	>100	1.05	70.4	<30	96	95	<0.8	None	None	Normal
**15**	>100	1.16	78.2	<30	97	94	<0.8	None	None	Normal
**16**	>100	1.10	75.5	<30	99	98	<0.8	None	None	Normal
**17**	>100	1.13	71.1	<30	98	94	<0.8	None	None	Normal
**18**	>100	1.13	82.5	<30	99	96	<0.8	None	None	Normal
**19**	>100	1.17	76.8	<30	98	96	<0.8	None	None	Normal
**20**	>100	1.28	72.2	<30	99	95	<0.8	None	None	Normal
**21**	>100	1.14	70.6	<30	99	96	<0.8	None	None	Normal
**22**	>100	1.05	75.5	<30	100	98	<0.8	None	None	Normal
**23**	>100	1.08	72.2	<30	99	96	<0.8	None	None	Normal
**24**	>100	1.12	72.9	<30	98	95	<0.8	None	None	Normal

Peak VO_2_% predicted: peak of Oxygen uptake expressed as percentage of the maximal predicted Oxygen Uptake; RER: Respiratory Exchange Ratio; 1st VT %VO_2_ peak: first Ventilatory Threshold Expressed as percentage of peak oxygen uptake; VE: Ventilation; VCO_2_: carbon dioxide uptake; SpO_2_: Peripheral Oxygen; MVV: Maximal Voluntary Ventilation.

**Table 3 jcm-10-03053-t003:** Comparison between COVID-19 athletes and Healthy Control.

	Covid-19 Athletes (*N* = 24)	HC (*N* = 11)	*p*-Value
Age	23.5 (20–25.5)	21 (10–24)	0.734
BMI kg/m	23.34 (22.67–24.16)	22.99 (22.30–25.65)	0.776
Asthma	3 (12.5)	0 (0)	0.220
Spirometry			
FVC L	5.415 (4.87–5.995)	5.46 (5.01–5.72)	0.696
FVC%	98 (93.5–105.5)	103 (100–104)	0.188
FEV1 L	4.38 (4.05–4.995)	4.8 (4.39–5.49)	0.078
FEV1% th	97.5 (91.5–108)	109 (106–116)	0.007
FEV1/FVC	83.55 (77.2–86.9)	88.7 (81.2–94.4)	0.166
FEV1/FVC% th	101 (91–105)	107 (98–114)	0·075
FEF25–75 L	4.665 (3.78–5.175)	4.72 (4.34–5.79)	0.384
FEF25–75% th	98 (78.5–108.5)	106 (94–130)	0.248
PEF% th	97 (86.5–108.5)	97 (92–120)	0.302
Echocardiography			
LVEDD mm	51 (50.15–53.35)	51 (50.2–54)	0.873
LVESDmm	32.3 (29.5–35.8)	33 (29–37.4)	0.683
IVSd mm	11 (11–11.65)	11 (10–12.0)	0.582
PWd mm	10.3 (10–11.0)	10 (10–11.0)	0.985
LVMass/BSA kg/m^2^	110.5 (107–115.5)	106 (97–114)	0.194
LVEDV mL	116 (106–123.5)	122 (117–125)	0.138
LVESV mL	41 (36.5–46)	48 (45–49)	0.683
EF%	63.5 (61–65)	62 (60–65)	0.225
TAPSE mm	23 (20.8–25)	21.4 (21–24)	0.519
PAP mmHg	24 (21–25.5)	18(17–22)	0.017
TAPSE/PAP	0.96 (0.84–1.14)	1.15 (0.95–1.33)	0.179
F.A.C%	50.75 (45.55–55.65)	52 (48.6–54.3)	0.582
PAacTime s	118 (113–127)	115 (112–123)	0.433
E/A	1.48 (1.32–1·68)	1.51 (1.20–1.75)	0.734
DT	169 (160–190)	174 (145–186)	0.929
Resting ECG			
PQ ms	165 (150–183)	154 (148–170)	0.569
QTc ms	408.5 (391–428.5)	402 (392–416)	0.396
Inc. RBBB	9 (37.5)	1 (9%)	0.084
Resting HR	63.5 (55–67)	62 (51–78)	0.887
CPET			
Peak HR	171 (166–179)	162 (155–183)	0.423
Resting SBP mmHg	120 (105–120)	120 (100–130)	0.749
Peak SBP mmHg	177.5 (170–182.5)	180 (170–190)	0.321
Resting SpO_2_%	98 (98–99)	98 (98–100)	0.458
Peak SpO_2_%	96 (95–96)	96 (95–97)	0.710
Peak VO_2_ mL/kg/min	50.1 (47.7–51.6)	49 (44.2–52.6)	0.618
Resting VE L/m	11.95 (9.5–13.5)	12.4 (9–14)	0.887
Peak VE L/m	106.8 (100.75–126.05)	115.4 (109.8–127.2)	0.319
Peak VCO_2_ L/min	3.818 (3.5215–4.194)	4.018 (3.899–4.396)	0.118
Peak RER	1.1 (1.105–1.14)	1.1 (1.08–1.18)	0.354
VE/VCO_2_ slope	27.35 (25.55–29.45)	28.1 (26.8–29.3)	0.271
Lowest VE/VCO_2_	23.3 (22.45–25.15)	25.1 (23.7–26.5)	0.155
1stVT VO_2_ mL/kg/min	36.45 (34.5–39.7)	38.2 (34.5–41.4)	0.789
1stVT% Peak VO_2_	74.07 (71.6–78.6)	79.04 (76.04–80.61)	0.082
Peak O_2_ pulse mL/kg/min	22.55 (20.55–25.1)	21.07 (19.5–22)	0.141

BMI: body mass index; HR: Heart Rate; SBP: Systolic Blood Pressure; DBP: Diastolic Blood Pressure; FVC: Forced Vital Capacity; FEV1: Forced Expiratory Volume in one second; FEF 25–75%: mean forced expiratory flow between 25% and 75%; PEF: Peak Expiratory Flow; LVEDD: Left Ventricular End Diastolic Diameter; LVESD: Left Ventricular End Systolic Diameter; IWSd: Intra ventricular Septal diameter; PWd: Posterior Wall Diameter; LVM: Left Ventricular Mass; BSA: Body Surface Area; LVEDV: Left Ventricle End Diastolic Volume; LVESV; Left Ventricle End Systolic Volume; EF: Ejection Fraction; TAPSE: tricuspid annular plane systolic excursion; PAP: systolic Pulmonary Artery Pressure; F.A.C: fractional area of change: DT: mitral deceleration time; Inc. RBBB: Incomplete Right Bundle Bunch Block VO_2_: oxygen uptake; VE: Ventilation; RER: Respiratory Exchange Ratio; VCO_2_: Carbon dioxide uptake.

## Data Availability

Data that underlie the results reported in this article, after deidentification, protocol and statistical analysis will be available on request. Researchers should provide a methodologically sound proposal.

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
