# Peer review of "Clinical Characteristics, Exercise Capacity and Pulmonary Function in Post-COVID-19 Competitive Athletes"

_jcm, 2021, doi:10.3390/jcm10143053_

Round 1

Reviewer 1 Report

In this report the clinical characteristics, exercise capacity
and pulmonary function were assessed in competitive athletes
shortly (no more than 30 days) after a positive rt-PCR SARS-CoV-2
RNA testing. The cases were compared with a group of athletes
with a negative test.

Concerns
1. The primary concern is the number of cases and controls
examined to extract useful conclusions. The statistical
power to identify rare incidences, in specific SARS-CoV-2
cardiac involvement ranges between 0.6-0.7%, as reported
(doi:10.1001/jamacardio.2021.0565 and doi:10.1161/CIRCULATIONAHA.121.054824).
2. The time between initial infection to cardiac complications
may exceeding the period post a positive test that the authors
examined.

Minor issue
1. There is a discrepancy in the abstract  "The most frequent
persisting symptoms were anosmia 145.83% and ageusia 8 33.33%."
instead of " The most frequent persisting symptoms were anosmia
(n=11;  162  45.83%) and ageusia (n=8; 33.33%)." in results.

Author Response

Reviewer 1

In this report the clinical characteristics, exercise capacity and pulmonary function were assessed in competitive athletes shortly (no more than 30 days) after a positive rt-PCR SARS-CoV-2
RNA testing. The cases were compared with a group of athletes
with a negative test.

Concerns
1. The primary concern is the number of cases and controls examined to extract useful conclusions. The statistical power to identify rare incidences, in specific SARS-CoV-2 cardiac involvement ranges between 0.6-0.7%, as reported (doi:10.1001/jamacardio.2021.0565 and doi:10.1161/CIRCULATIONAHA.121.054824).

Reply:During the last six months some information regarding cardiac abnormalities in post-Covid-19 athletes has been published. Multicenter studies as Daniels et al 2021  doi:10.1001/jamacardio.2021.2065, and Moulsen et al 2021included a very high number of athletes, and based on CMR imaging evaluation reported an overall prevalence of myocarditis of 2.3% and 0.5% respectively. It should be mentioned that the limited number of participants is a limitation of our study, but other previous and recent reports as Vago et al ( 12 subjects) doi:10.1016/j.jcmg.2020.11.014 , Malek et al 2021 (26 subjects) doi:10.1002/jmri.27513 ,  Clark et al 2021 (59 subjects)  doi:10.1161/CIRCULATIONAHA.120.052573  , also reported  low prevalence of cardiac involvement, in line with the results of multicentered studies. Studies with huge number of participants described variability of myocarditis prevalence across different centers, indicating first of all the necessity of standardized protocols regarding clinical screening and standardized CMR protocols interpretation, and second but not less important the necessity of further data and information. Therefore, we believe that our data based on clinical and instrumental evaluations of post-Covid-19 athletes are of clinical interest.  To clarify this point we added the following text in introduction section: ''At present several information regarding myocarditis in young athletes after COVID-19 disease has been described with heterogenous prevalence ranging from 0-15%, indicating the necessity of further data and standardized screening protocols before a safe return to play''. and the following text in discussion: ''''.. and recently data from a multicentered prospective study reported that the prevalence of cardiac involvement following SARS-CoV-2 infection is about 0.5-3% and no adverse cardiac events were registered during a median follow-up of 113 days in young competitive athletes.'' 

  1. The time between initial infection to cardiac complications may exceeding the period post a positive test that the authors examined.

Reply:During the last months the progress of articles bringing light on COVID-19 consequences is remarkable, however, to the best of our knowledge follow-up data on cardiac events in post-Covid-19 athletes are reported only by Moulsen et al. 2021 where during a mean follow up of about three months no events were registered. Follow up data could not be the aim of this study, given limited time from enrollment (almost all patients will reach at least 6 months follow up during July of 2021). Secondly from our baseline clinical evaluations and CPET monitoring we did not reveal complex arrythmias or other clinical conditions suggestive for cardio-respiratory implications for the moment. However, considering that viral myocarditis has been documented as a cause of sudden cardiac death among young athletes, Peterson et al 2020 doi: 10.1136/bjsports-2020-102666and more in than 50 % of cases no suggestive symptoms as chest pain, dyspnea or palpitations were present, we agree with the Reviewer, therefor we added the following sentences in discussion and study limitations: 

''Other studies evaluating symptoms persistence in adults with mild COVID-19 disease, reported that during three to six months follow up palpitations, chest pain and dyspnea were present respectively in 9, 5 and 30% of adults [35]. Another study including 1946 patients who suffered out of hospital cardiac arrest and 1080 in hospital cardiac arrest had a 2.3 to 3.4-fold risk of dying within one month and COVID-19 was involved in at least 10 % of out of hospital cardiac arrest [36]. Post-mortem studies of athletes who died from sudden cardiac death myocarditis was diagnosed in up to 8% [37]. Moreover, only in 21.6% of sudden cardiac deaths related to myocarditis among young athletes was characterized by typical symptoms as chest pain, dyspnea and palpitations [38]. Therefore, short and long-term outcome data are strongly required to clarify the implications of COVID-19 disease on cardiac events''.

''Finally, in this study we did not provide follow-up data regarding cardio-respiratory consequences of SARS-CoV-19 infection in competitive athletes, and long-term implications could not be excluded, however baseline examinations and CPET monitoring excluded the presence of complex arrythmias or other conditions of clinical significance''

Minor issue

  1. There is a discrepancy in the abstract  "The most frequent persisting symptoms were anosmia 145.83% and ageusia 8 33.33%." instead of " The most frequent persisting symptoms were anosmia
    (n=11;  162  45.83%) and ageusia (n=8; 33.33%)." in results.

Reply:We apologies for the typing error, this was corrected.

We wish to thank the Reviewer for the suggestions and constructive criticism and hope that after the revision process  satisfied His/Her requests and the Reviewer finds our manuscript ameliorated.

Reviewer 2 Report

Interesting paper, thank you

Abstract, not sure how a symptom can be > 100%, is this what you meant to say?

Page 2 Inclusion criteria: upper age limit?
disease was asymptomatic to mild,  anybody get hypoxic in this group.Very different outcomes in patients with more severe covid illness, 
Point may be that these healthy young athletes did not get severe covid, which would be what we predicted....

Page 3:

spirometry recommended against by all socieities post covid, how did you get around this?

Page 13, FEv1 of theoretical significance, you do expand a bit on this later, but need to relate to an MCID or clinical significance

Page 14, your comment on steroids and mortality: 

what does this treatment for severe disease have to do with your discussion of young people with mild disease? I would remove this

Conclusion: "plausible protective role .." :

what evidence do you use to create this hypothesis? Might it just be a young healthy non overweight cohort effect?

Author Response

Reviewer 2

Interesting paper, thank you

Abstract, not sure how a symptom can be > 100%, is this what you meant to say?

Reply:We apologies, it was a typing error the correct sentence is: The most frequent persisting symptoms were anosmia 11 (45.83%) and ageusia 8 (33.33%).

Page 2 Inclusion criteria: upper age limit?

disease was asymptomatic to mild,  anybody get hypoxic in this group.Very different outcomes in patients with more severe covid illness,

Point may be that these healthy young athletes did not get severe covid, which would be what we predicted....

Reply: We did not study competitive athletes over 35 years old.In the revised version of our manuscript this information is now reported on method section.

None of our patients referred hypoxia and none of them required hospitalization. We strongly agree with the Reviewer's suggestion regarding the mild COVID-19 presentation and the lack of consequences regarding physical performance or respiratory impairment. Indeed, a recent study Guler SA et al 2021 https//doi. org/10.1183/13993003.03690-2020, focused on the pulmonary sequelae of COVID-19 in patients which experienced mild, moderate or severe/ critical disease after 4 months of follow-up  reveled that patients with severe or critical illness presented reduced pulmonary function es measured by DLCO and oxygen desaturation on exercise and reduced distance in 6MWT. Despite some other recent few reports, Huang et al 2021 doi: 10.1016/S0140-6736(20)32656-8; Xu et al 2021 doi: 10.1016/S2213-2600(21)00174-0. the evidence still remains uncertain, and obviously more information regarding exercise capacity and follow up are needed.

Following the interesting point discussed by the Reviewer we modified the text in introduction section and also added the following text in discussion section:

''Despite limited evidence regarding COVID-19 sequelae on respiratory and physical function, few reports suggest impairment during short term follow-up and a recent study described that patients which experienced severe to critical COVID-19 disease are characterized by reduction of pulmonary function as measured by diffusing capacity of the lung for carbon monoxide (DLCO), radiological findings of ground glass opacities with mosaic attenuation, up to four months of follow-up, which can lead to ventilation/perfusion mismatching and contribute to physical impairment and hypoxemia during exercise [27]'' .

Page 3:

spirometry recommended against by all socieities post covid, how did you get around this?

Reply: According to Italian Federation of Sport Medicine (FMSI), evaluation of pulmonary function by spirometry is recommended in all athletes (post-COVID -19 or COVID-19 negative) in terms of eligibility to perform agonistic physical activity. The protocol is available in Italian language however you can check here: https://fmsi.it/images/img/news/Circolare-idoneit-sportiva-np-covid-13-1-21.pdf.

Page 13, FEv1 of theoretical significance, you do expand a bit on this later, but need to relate to an MCID or clinical significance

Reply: Data regarding minimal clinically important difference in pulmonary parameters in post-Covid-19 are lacking and future studies are needed to explore if these variations have a clinical significance. Therefor we made a comparison with asthmatic population where in short term trials FEV1 changes of 20% are of clinical importance. In addition, some recent studies also report reduction of spirometry parameters during short term follow-up and up to 12 months in particular in patients after moderate or severe COVID-19. (please check below the study by Guner and Xu et al).

Page 14, your comment on steroids and mortality:

what does this treatment for severe disease have to do with your discussion of young people with mild disease? I would remove this

Reply:In this point we wanted to suggest that as exercise modulates hormones release, probably subjects who exercise intensively have a different inflammatory response related to cortisol release and better outcome. As this may be confounding, this was removed.  

Conclusion: "plausible protective role .." :

what evidence do you use to create this hypothesis? Might it just be a young healthy non overweight cohort effect?

Reply:Obesity has emerged as a risk factor in COVID-19 severity and outcome Stefan et al  doi:10.1038/s41574-020-0364-6 and probably physical exercise plays a crucial role in this. As we did not compare athletes vs non athletes we modified the conclusion, by removing the part of protective role of exercise and adding a sentence in discussion.  

Thank You for reviewing our manuscript, and for the constructive suggestions!

Reviewer 3 Report

Comments to Authors:

The authors retrospectively to evaluate the clinical presentation of COVID-19 in a cohort of competitive athletes as well as spirometry and echocardiography findings and cardio-respiratory performance during exercise. They found that reduced exercise capacity was not identified and pulmonary and 39 cardiovascular function are not impaired during early recovery phase in a population of physical 40 active adults except FEV1 reduction.

I have the following concerns.

Comment 1

Table 3

Asma is miss-spelled.

Comment 2

In this athlete's study, most cases had mild symptoms. The effects on the body after onset were minor. What is your vison in the future from this result?

Comment 3

There are currently multiple treatments for COVID-19, did these athletes receive any treatment?

Author Response

Reviewer 3

The authors retrospectively to evaluate the clinical presentation of COVID-19 in a cohort of competitive athletes as well as spirometry and echocardiography findings and cardio-respiratory performance during exercise. They found that reduced exercise capacity was not identified and pulmonary and 39 cardiovascular function are not impaired during early recovery phase in a population of physical 40 active adults except FEV1 reduction.

I have the following concerns.

Comment 1

Table 3

Asma is miss-spelled.

 Reply: Thank you! We corrected this.

Comment 2

In this athlete's study, most cases had mild symptoms. The effects on the body after onset were minor. What is your vison in the future from this result?

Reply:Since the trajectory of post-COVID -19 is still unknown we plan to complete a strict short term and long term follow-up evaluation in our patients, including complete echocardiography, CPET and spirometry evaluations, and also evaluation of persisting symptoms.

Comment 3

There are currently multiple treatments for COVID-19, did these athletes receive any treatment?

Reply:Athletes with mild symptoms as fever, slight cough and myalgia received non-steroidal anti-inflammatory drugs during symptoms presentation. Asymptomatic athletes or athletes presenting myalgia, fatigue but no fever didn't take any treatment except isolation and rest.

Thank You for the comments, constructive suggestions and for reviewing our paper.   

Reviewer 4 Report

good work

Author Response

Reviewer 4

good work

Thank You for the comments, for appreciating and reviewing our article!

Round 2

Reviewer 1 Report

In this report the clinical characteristics, exercise capacity
and pulmonary function were assessed in competitive athletes
shortly (no more than 30 days) after a positive rt-PCR SARS-CoV-2
RNA testing. The cases were compared with a group of athletes
with a negative test.

Author Response

Thank You for Reviewing. English language was checked.

Reviewer 3 Report

This article has been improved.

Asma is miss-spelled in table 1 and 3.

I have no additional requests for content.

Author Response

Thank you for the suggestion,  we corrected in table 1 and table 3 ''asthma'' instead of ''asma''